# Comparative fertility and pregnancy outcomes after local treatment for cervical intraepithelial neoplasia and stage 1a1 cervical cancer: protocol for a systematic review and network meta-analysis from the CIRCLE group

Antonios Athanasiou,[1,2] Areti Angeliki Veroniki,[1,3] Orestis Efthimiou,[4] Ilkka Kalliala,[1,5] Huseyin Naci ![ORCID],[6] Sarah Bowden,[1,2] Maria Paraskevaidi,[1] Pierre Martin-Hirsch,[7] Philip Bennett,[1,2] Evangelos Paraskevaidis,[2,8] Georgia Salanti,[4] Maria Kyrgiou[1,2]

For numbered affiliations see end of article.

**Correspondence to**
Dr Maria Kyrgiou;
m.kyrgiou@imperial.ac.uk

## ABSTRACT

**Introduction** There are several local treatment methods for cervical intraepithelial neoplasia that remove or ablate a cone-shaped part of the uterine cervix. There is evidence to suggest that these increase the risk of preterm birth (PTB) and that this is higher for techniques that remove larger parts of the cervix, although the data are conflicting. We present a protocol for a systematic review and network meta-analysis (NMA) that will update the evidence and compare all treatments in terms of fertility and pregnancy complications.

**Methods and analysis** We will search electronic databases (CENTRAL, MEDLINE, EMBASE) from inception till October 2019, in order to identify randomised controlled trials (RCTs) and cohort studies comparing the fertility and pregnancy outcomes among different excisional and ablative treatment techniques and/or to untreated controls. The primary outcome will be PTB (<37 weeks). Secondary outcomes will include severe or extreme PTB, prelabour rupture of membranes, low birth weight (<2500 g), neonatal intensive care unit admission, perinatal mortality, total pregnancy rates, first and second trimester miscarriage. We will search for published and unpublished studies in electronic databases, trial registries and we will hand-search references of published papers. We will assess the risk of bias in RCTs and cohort studies using tools developed by the Cochrane collaboration. Two investigators will independently assess the eligibility, abstract the data and assess the risk of bias of the identified studies. For each outcome, we will perform a meta-analysis for each treatment comparison and an NMA once the transitivity assumption holds, using the OR for dichotomous data. We will use CINeMA (Confidence in Network meta-analysis) to assess the quality of the evidence for the primary outcome.

**Ethics and dissemination** Ethical approval is not required. Results will be disseminated to academic beneficiaries, medical practitioners, patients and the public.

**PROSPERO registration number** CRD42018115495

## Strengths and limitations of this study

► This will be the first network meta-analysis (NMA) to produce comprehensive summaries of the relative reproductive morbidity of treatment methods for cervical preinvasive and early invasive disease.

► We will use state-of-the-art methods for combining randomised and non-randomised studies in an NMA.

► Risk of bias will be evaluated at both study and outcome level.

► One possible limitation of this review is that we expect to find mainly retrospective cohort studies at high risk of recall, selection and publication bias.

## INTRODUCTION

The introduction of systematic call and recall screening programmes in the UK has resulted in a profound decrease in the incidence and mortality from cervical cancer, as preinvasive precursors (cervical intraepithelial neoplasia; CIN) can be detected by the screening programme and treated.[1] In England alone, 3.6 million women aged 25–64 years attended for screening in 2013–2014 and over 23 800 treatment procedures were carried out.[2]

Local conservative treatment for cervical preinvasive and early invasive disease removes or ablates a cone-shaped part of the cervix containing the precancerous cells. The choice of technique varies within the UK, across Europe and beyond. In some countries, knife excision (cold knife conisation; CKC) is still regularly performed; in others, laser ablation or laser conisation with the laser beam is common practice. In the UK, large loop excision of the transformation zone (LLETZ) is the preferred treatment, with some units

offering alternative techniques more frequently than others. This preference is because LLETZ is quick, easy to do and of low cost.

The mean age of women undergoing CIN treatment is similar to the age of women having their first child. Although, previously, complications from treatment were thought to be relatively mild and uncommon, an increasing body of retrospective observational studies and meta-analyses suggested that treatment, particularly excision, adversely affects future reproduction and the risk of prematurity.[3–8] It has been suggested that the frequency and severity of the observed adverse events is higher for the more radical techniques and with increasing cone depth.[4 7 9–14] Preterm birth (PTB) is a major cause of neonatal death and disability and represents an enormous cost to the health services and the society in general.

Although all treatment techniques are highly effective in preventing recurrent precancerous disease and future invasion,[15] some of the data on the risk of reproductive morbidity for the treatment methods has been conflicting[3 5 7 10 16–18] and their comparative reproductive morbidity remains unclear. With some authors raising concerns that the progressive reduction in the radicality of treatment has led to increased risk of future post-treatment invasive disease,[19 20] and others advocating the move to less radical techniques, such as laser ablation (LA), for the prevention of treatment-associated adverse obstetric outcomes, such as PTB or perinatal mortality,[4 21] high-quality synthesis of the current evidence base is an urgent unmet need. Given the premalignant nature of the condition, a randomised controlled trial (RCT) comparing the various treatment technique to no treatment will never be conducted.

The quantification of the comparative reproductive morbidity of different treatment techniques and cone lengths has become a women's health priority. This requires high-quality synthesis of the evidence in comprehensive summaries that will become available for effective patient counselling at colposcopy and antenatal clinics for patients, clinicians and policymakers. A clinical ranking of treatments with regards to the risk of PTB may allow the quantification of risk and the detection of women at high risk of PTB that would benefit from intensive surveillance antenatally, while minimising the unnecessary interventions for those at lower risk.

The key methodological vehicle to synthesise evidence is systematic reviews and their quantitative component, meta-analysis. Network meta-analysis (NMA) is an extension of pairwise meta-analysis, which can be used to estimate the relative effectiveness of several competing treatments. An NMA has never been used before to assess the comparative efficacy and harms of the different treatment techniques in this field. For every treatment comparison NMA synthesises both direct evidence (ie, coming from studies comparing head-to-head the treatments of interest) and indirect evidence (ie, coming from studies comparing the treatments of interest via an intermediate common comparator).[22–25] In addition, NMA allows the estimation of relative effects between all available treatments, can lead to an increased precision as compared with the pairwise meta-analysis and provides a ranking of the available competing treatments. The potential of NMA has been recognised by the National Institute of Clinical Excellence[26] and several international Health Technology Assessment agencies.[27 28]

The objective of this systematic review and NMA is to update the evidence and to compare the various local treatment methods to manage CIN in terms of fertility, early (<24 weeks of gestation) and late (>24 weeks) pregnancy complications. This is part of the CIRCLE project (Cervical Cancer Incidence, CIN Recurrence and Reproduction after Local Excision), which aims to generate a clinically useful raking of alternative options for treatment of CIN according to their efficacy (risk of preinvasive and invasive recurrence), morbidity and cost-effectiveness.

## METHODS AND ANALYSIS
This protocol complies with the Preferred Reporting Items for Systematic Review and Meta-Analysis Protocols (online supplementary file 1).[29] Any changes in this protocol will be recorded in an updated version of the PROSPERO registration.

### Criteria for considering studies for this review
#### Types of participants
We will include women of all ages with a prior history of local surgical treatment for CIN or microinvasive early cervical cancer (stage IA1). Their status can be confirmed with histological or cytological diagnosis and irrespective of the grade of the treated lesion for both squamous and glandular intraepithelial neoplasia. Studies recruiting solely women at high risk of PTB (such as previous history of PTB) or studies including only patients treated during pregnancy will be excluded.

#### Types of interventions
We aim to compare nine different excisional or ablative techniques used for conservative treatment for CIN. The excisional techniques include CKC; laser conisation; needle excision of the transformation zone, also known as straight wire excision of the transformation zone; LLETZ, also known as loop electrosurgical excisional procedure; Fischer cone biopsy excision. The ablative techniques include radical point diathermy; cryotherapy; cold coagulation; LA. We will include studies comparing any of these treatments with each other or with no treatment. Figure 1 shows a network example of all possible comparisons between eligible interventions for the primary analysis. If the specific treatment technique is not specified, these will be grouped under the broader categories excision or ablation.

#### Outcome measures
##### Primary outcome
1. PTB, defined as <37 weeks of gestation.

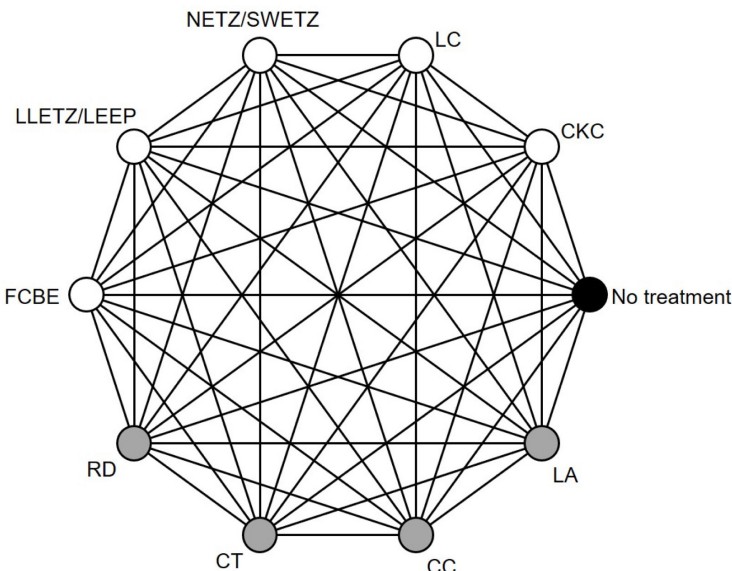

**Figure 1** Network of possible pairwise comparisons between eligible treatment methods. CC, cold coagulation; CKC, cold knife conisation; CT, cryotherapy; FCBE, Fischer conebiopsy excision; LA, laser ablation; LC, laser conisation; LLETZ, large loop excision of the transformation zone, also known as LEEP, loop electrosurgical excisional procedure; NETZ, needle excision of the transformation, also known as SWETZ, straight wire excision of the transformation zone; RD, radical point diathermy.

### Secondary outcomes

1. Spontaneous PTB, defined as <37 weeks of gestation.
2. Severe PTB, defined as <32/34 weeks of gestation.
3. Extreme PTB, defined as <28/30 weeks of gestation.
4. Prelabour rupture of membranes defined as membrane rupture before the onset of labour.
5. Low birth weight, defined as infant born weighing <2500 g (late neonatal obstetric outcome).
6. Neonatal intensive care unit admission.
7. Perinatal mortality, defined as the number of stillbirths and neonatal deaths occurring within 28 days after birth.
8. Total pregnancy rate, defined as any pregnancy occurring from CIN treatment till study completion irrespective of outcome (miscarriage, ectopic, molar pregnancy, termination of pregnancy, live birth, stillbirth).
9. Rates of women requiring >12 months to conceive.
10. First trimester miscarriage, defined as miscarriage at <12 weeks of gestation.
11. Second trimester miscarriage, defined as miscarriage between 12 and 24 weeks of gestation.

Outcomes and their classification into primary or secondary were decided after clinical experts' opinion. Total pregnancy rate will be recorded for the whole study period and/or prespecified intervals, if data are available.

### Types of studies

We will include RCTs, quasi-RCTs and cohort studies comparing fertility and early (<24 weeks of gestation) or late (>24 weeks of gestation) pregnancy outcomes among surgical techniques and those that compare to no treatment. Studies that do not perform a comparison between treatments (ie, 'single-arm studies') will

be excluded. Studies that compared a treatment with an untreated group will be included irrespective of the type of the untreated group (eg, studies that used data from untreated women from the general population; studies that used self-matching, ie, including women with pregnancies before and after treatment; studies that used data from women with a history of abnormal cytology/HPV infection/untreated CIN). There will be no time or language restriction.

### Information sources and search strategy

An experienced librarian will search The Cochrane Gynaecological Cancer Specialised Register; Cochrane Central Register of Controlled Trials (CENTRAL); MEDLINE; and EMBASE for eligible studies from inception. The search algorithms for these databases are presented in online supplementary file 2. We will search Metaregister, Physicians Data Query, www.controlled-trials.com/rct, www.clinicaltrials.gov and www.cancer.gov/clinicaltrials for ongoing studies. There will be no time or language restriction.

We will contact the corresponding author of any relevant ongoing trials for further information and unpublished data. In an attempt to identify any articles missed by the initial search, we will use the 'related articles' feature in MEDLINE. We will also hand search the references of the retrieved articles and meta-analyses. We will search conference proceedings and abstracts through ZETOC (http://zetoc.mimas.ac.uk), and theses through WorldCat Dissertations. The selected conferences will include: British Society of Colposcopy and Cervical Pathology; International Federation of Cervical Pathology and Colposcopy; Annual Meeting of European Federation of Colposcopy; Annual Meeting of the American Society of Colposcopy

and Cervical Pathology. We will contact experts in the field, including directors of UK cancer and colposcopy registries, to identify further reports of studies. We will include both published and unpublished studies.

### Study selection

We will download abstracts retrieved into a reference management software, Zotero. Then, two persons will independently review titles and abstracts retrieved by the search (level 1). At level 2, we will obtain the full text of all included articles and two reviewers will independently use the same inclusion criteria to determine eligibility. Disagreements at any level will be resolved via discussion with a third member of the review team.

### Data collection

Two reviewers will extract data independently using a standardised data collection form in Excel. Disagreements will be resolved through discussion. Information extracted will include study characteristics (such as author, publication year and study design), participants (such as age, CIN grade and smoking) and comparison group characteristics, setting, inclusion/exclusion criteria, intervention details, outcome measures and dropout rates. In RCTs, we will prefer arm-level data (number of events and sample size per intervention arm), but if these are missing, the study-level data will be used in the analysis, for example, reported ORs and a measure of their uncertainty (eg, CI). In observational studies, we will extract estimates of treatment effects that are adjusted for the lack of randomisation, that is, after taking into account the impact of potential confounders, or, if these are missing, the reported unadjusted estimates, as well as the corresponding uncertainty measure.

### Risk of bias assessment

Results of the meta-analyses will be interpreted in light of risk of bias assessment of the included studies. Two investigators will independently assess the methodological quality/risk of bias of the studies that fulfil the eligibility criteria and differences will be resolved by discussion with a third investigator.

For RCTs, the risk of bias will be assessed using the tool[30] developed by Cochrane assessing the following domains: randomisation process, deviations from the intended interventions, missing outcome data, measurement of the outcome and selection of the reported result. The risk of bias in each domain, as well as the overall risk of bias, will be rated as 'low risk', 'some concerns' or 'high risk', after answering the signalling questions of each domain with 'Yes', 'Potentially Yes', 'Potentially No' or 'No'. When inadequate detail s are reported in the study to be able to rate a risk of bias item, we will contact the study authors for additional information.

For non-randomised studies (NRS), we will use the ROBINS-I tool[31] developed by the Cochrane collaboration that facilitates the evaluation of the risk of bias by considering that each NRS is an attempt to mimic an

RCT comparing the effects of the intervention or exposure studied. During the review stage, we will evaluate the risk of bias in the following domains: confounding, selection of participants into the study, classification of interventions, deviations from intended interventions, missing data, measurement of outcomes and selection of the reported results. The confounding factors that we will evaluate are age, parity and smoking. Each ROBINS-I domain and the overall risk of bias will be assessed as 'low', 'moderate', 'serious' or 'critical', after answering the signalling questions of each domain with Yes, Potentially Yes, Potentially No or No.

### Statistical synthesis

#### Characteristics of included studies and network

We will generate descriptive statistics for eligible studies and study population characteristics, describing the types of comparisons and important clinical or methodological variables (such as publication year, study design and source of data). We will present the evidence in a network diagram per outcome. The total number of patients will be reflected in the size of the nodes, while the weight of each edge will be proportional to the number of studies per treatment comparison.

#### Pairwise meta-analyses

We will synthesise data to obtain summary ORs for dichotomous outcomes with a 95% CI in an inverse variance random-effects model assuming that the studies are estimating different but related treatment effects. In each meta-analysis, we will estimate the between-study variance with the restricted maximum likelihood estimator[32 33] and its 95% CI using the Q-profile approach.[34] We will assess between-study variance using the $I^2$ statistic along a 95% CI.[35 36] We will estimate each summary effect size and its 95% CI using the Hartung-Knapp-Sidik-Jonkman method[32 37 38] to handle meta-analyses that include a small number of studies. All meta-analyses will be conducted in R[39] using the *metafor* package.[40]

#### Network meta-analyses

*Data synthesis*

We will fit a random effects NMA model, taking into account the correlation induced by multiarm studies.[41] We will use a random-effects model, since we anticipate methodological and clinical heterogeneity across studies. We will assume a common between-study variance parameter for all comparisons in the network, so that treatment comparisons informed by a single study can borrow strength from the remaining network.[42 43] Clinically, this assumption is reasonable because all treatments included in the network are of the same nature. We will estimate the common between-study variance with the DerSimonian and Laird method of moments approach.[44]

We expect that we will include several NRS in our dataset. In that case, we will employ the methods described by Efthimiou *et al*,[45] that is, a 'design-adjusted', and a three-level hierarchical NMA model. Using these methods, we

will incorporate the totality of available information, both randomised and observational, in a joint NMA. We will start by analysing the study-specific estimates from the NRS at face value. Then, in extensive sensitivity analyses, we will explore the impact of assigning different levels of credibility and subsequently down-weighting the NRS, according to experts' opinion and the risk of bias as assessed in ROBINS-I.

For each treatment comparison, we will report the estimated OR, the 95% CI and the 95% prediction intervals. We will also estimate the ranking probabilities for all treatments of being at each possible rank for each intervention. Thus, we will obtain a treatment hierarchy using the surface under the cumulative ranking curve (SUCRA) or P-scores and mean ranks.[46] A rank-heat plot will be used to depict the SUCRA values or P-scores for all outcomes.[47 48] All NMAs will be fit in R[39] with the *netmeta*[49] and *rjags*[50] package.

We will repeat our NMAs after grouping all excisional and all ablative techniques together; see groupings of treatments in figure 1.

### Assessment of the transitivity assumption

NMA rests on the assumption of transitivity, that is, that effect modifiers have a similar distribution across treatment comparisons in a network.[24 51 52] In order to assess the plausibility of this assumption, we will summarise study and patient-level characteristics that are expected to influence relative treatment effects, for each pairwise comparison for which direct evidence is available in the network. In this NMA, the most important effect modifiers are expected to be year of study, method of ascertainment of exposure/outcome (hospital records, registries or interviews/questionnaires), age, parity, smoking and CIN grade. We will visually inspect the similarity of the identified studies in terms of these effect modifiers. We will investigate the inclusion and exclusion criteria of all studies, to make sure that patients, treatments and outcomes in the studies are sufficiently similar in all aspects that are expected to modify relative treatment effects.

### Assessment of statistical inconsistency

Checking the network for inconsistency offers an additional way of assessing the validity of the transitivity assumption. In order to evaluate the presence of inconsistency locally, we will separate the indirect from the direct evidence for each comparison and infer about their differences following the back-calculation method.[53] We will also follow a global approach for assessing consistency in the network, by applying the design-by-treatment interaction model.[54] Simulations suggest that inconsistency tests have low power to detect true inconsistency.[55 56] Therefore, we will conceptually assess the transitivity assumption (see previous paragraph) even in the absence of evidence for inconsistency. We will perform both local

(back-calculation method) and global (design-by-treatment interaction model) assessments in R[39] using the *netmeta* package.[49]

When a network includes both randomised and non-randomised evidence, we will also explore differences between the different types of evidence, as discussed in Efthimiou *et al*.[45] For each treatment comparison, there may be up to four different types of evidence: direct randomised, indirect randomised, direct non-randomised and indirect non-randomised. We will summarise all evidence by type, for each treatment comparison. If data permits, important discrepancies between these types will be further investigated, as they might indicate a breach of the transitivity assumption (eg, when randomised and non-randomised evidence are very different in terms of populations, interventions and so on), or the presence of important, unaccounted confounding in the non-randomised evidence. If a source of disagreement is identified, it will be included in our analysis through network meta-regression models.[23]

### Exploring heterogeneity and inconsistency: subgroup analyses, meta-regression and sensitivity analyses

We will assess the extent of statistical heterogeneity by inspecting the 95% prediction intervals and by comparing the estimated value of the between-study variance with the empirical distribution derived by Turner *et al* for dichotomous data.[57] For the primary outcome, we will explore the following possible sources of heterogeneity and inconsistency: year of study, method of ascertainment of exposure/outcome (hospital records, registries or interviews/questionnaires), age, parity, smoking and CIN grade. If sufficient studies are available, the role of these variables will be explored by means of subgroup analyses (categorical characteristics) or network meta-regressions (continuous characteristics).

### Reporting bias and small-study effects

In order to assess possible existence of small studies giving different effect estimates than larger studies, we will visually explore the funnel-plots for each treatment against the untreated group (using the relevant studies) when at least 10 studies inform the underling treatment comparison. We will also assess for small-study effects using the comparison adjusted funnel plot[58] and will conduct a network meta-regression using the study variance as a covariate.[59 60]

### Assessment of the credibility of the evidence

We will evaluate the credibility of the evidence contributing to the network estimates in the primary outcome using CINeMA[61] (http://cinema.ispm.ch/). Two team members will determine the degree of confidence in the estimated NMA results by assessing the six CINeMA domains: within-study bias (ie, risk of bias in the included studies), across-study bias (ie, publication and reporting bias), indirectness, imprecision, heterogeneity and incoherence (ie, differences between direct and indirect

evidence).[61] Each network summary estimate will initially be judged as high quality, but it will be downgraded if this is judged appropriate according to the six domains. Judgements within each domain will be summarised for each NMA relative treatment effect as: very low, low, moderate or high.

## Patient and public involvement

We have discussed the project with Jo's Cervical Cancer Trust (a UK charity who supports patients affected by cervical preinvasive or invasive cervical disease and campaigns for excellence in cervical cancer treatment and prevention).

Two patient representatives through Jo's Trust with personal experience of cervical disease have assisted us in the design of the study and the development of research questions. We aim to recruit more patients through Jo's Voice, who will help us understand the key priorities from patients' perspective, produce lay summaries and disseminate the results to the wider public.

## Ethics and dissemination

This review does not require ethical approval. We identified four groups of potential stakeholders (academic beneficiaries; health-related agencies and decision-makers; medical practitioners; patients and public) and specific action items to effectively target them. We will publish papers in influential open access journals and we will present data at high-profile conferences. We will make the datasets available to the wider research community. We will organise a workshop with key stakeholders. We will develop information sheets and briefings, highlighting the key findings and circulate newsletters. We will engage the press with presentations and social media interviews and we will work closely with Jo's Trust charity that plays an important role in educating patient communities.

**Author affiliations**
[1]Department of Surgery and Cancer, Faculty of Medicine, Imperial College London, Institute of Reproductive and Developmental Biology, London, UK
[2]West London Gynaecological Cancer Centre, Imperial College Healthcare NHS Trust, London, UK
[3]School of Education, Department of Primary Education, Panepistimio Ioanninon, Ioannina, Greece
[4]Institute of Social and Preventive Medicine (ISPM), University of Bern, Bern, Switzerland
[5]Department of Obstetrics and Gynaecology, University of Helsinki and Helsinki University Hospital, Helsinki, Finland
[6]Department of Health Policy, London School of Economics, London, UK
[7]Department of Gynaecologic Oncology, Lancashire Teaching Hospitals NHS Foundation Trust, Preston, UK
[8]Department of Obstetrics and Gynaecology, University of Ioannina and University Hospital of Ioannina, Ioannina, Greece

**Acknowledgements** We thank the Jo's Trust for their assistance to involve public and patient representatives in this project.

**Contributors** The study was conceived and designed by MK, GS and EP. The protocol was drafted by AA, MK, AAV, OE, IK, GS and was revised critically for important intellectual content by all authors (AA, AAV, OE, IK, HN, SB, MP, PM-H, PB, EP, GS, MK). MK is the guarantor.

**Funding** This work is supported by National Institute for Health Research (NIHR) Research for Patient Benefits (P67307) (MK). MK is also supported by the British Society of Colposcopy Cervical Pathology Jordan/Singer Award (P47773) and the Imperial College Healthcare Charity (P47907) (MK). AAV is also supported by the European Union's Horizon 2020 (No 754936). None of the funders have any influence on the study design; in the collection, analysis and interpretation of data; in the writing of the report; and in the decision to submit the article for publication.

**Competing interests** None declared.

**Patient consent for publication** Not required.

**Provenance and peer review** Not commissioned; externally peer reviewed.

**ORCID iD**
Huseyin Naci http://orcid.org/0000-0002-7192-5751

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
