## [Reviewer comments · BMJ Open]

ARTICLE DETAILS

TITLE (PROVISIONAL)	Comparative fertility and pregnancy outcomes after local treatment for cervical intra-epithelial neoplasia and stage 1a1 cervical cancer: protocol for a systematic review and network meta-analysis from the CIRCLE Group
AUTHORS	Athanasiou, Antonios; Veroniki, Areti Angeliki; Efthimiou, Orestis; Kalliala, Ilkka; Naci, Huseyin; Bowden, Sarah; Paraskevaidi, Maria; Martin-Hirsch, Pierre; Bennett, Philip; Paraskevaidis, Evangelos; Salanti, Georgia; Kyrgiou, Maria

VERSION 1 – REVIEW

REVIEWER	Dr Aime Powell University of Notre Dame Australia (Fremantle), Western Australia, Australia.
REVIEW RETURNED	04-Feb-2019

GENERAL COMMENTS	Manuscript title: Comparative fertility and pregnancy outcomes after local treatment for cervical intra-epithelial neoplasia and stage 1a1 cervical cancer: a protocol for a systematic review and network meta-analysis from the CIRCLE group. Abstract - Very well written and presents a clear summary of the proposed work to be undertaken. Article Summary - details strengths and limitations clearly. Consideration has been given to the fact that mainly retrospective cohort studies will be identified. However, given the malignancy risks to patients an RCT comparing treatment techniques will not be conducted. Introduction - provides clear and concise background information. The proposed study is novel as it is a meta-analysis and network meta-analysis which compare fertility treatments for women with CIN and early stage cervical cancer. Methods and analysis: - Objectives are clearly defined and of clinical interest to identify ways to improve patient outcomes.- Information sources and search strategy is robust and extensive.- Risk of bias is well considered and appropriate tools referenced.- Statistics/Analysis are logical and comprehensive. No requirement for an external biostatistical review.- Consumer involvement noted, this is an excellent addition to the manuscript.
--

	Minor comment: Within the papers searched will you be able to determine if the patient had undergone reproductive technology prior to their CIN or early stage cervical cancer being treated?
REVIEWER	N. Danhof Amsterdam UMC, The Netherlands
REVIEW RETURNED	05-Apr-2019
GENERAL COMMENTS	This manuscript represents a study protocol of a network meta analysis on local treatment for CIN, which is an important topic. To the best of my knowledge, I am not aware of a network meta analysis performed on this topic. The problem and the methods are well addressed. The authors can maybe explain the fertility outcomes in more detail. In the secondary outcomes they have described total pregnancy rate. How is "pregnancy" defined? How long is the follow up time? Do they extract data on time to pregnancy? Did the authors consider to differentiate between spontaneous preterm delivery and iatrogenic preterm delivery, for example in case of pre eclampsia?

VERSION 1 – AUTHOR RESPONSE

Reviewer: 1

Reviewer Name: Dr Aime Powell

Institution and Country: University of Notre Dame Australia (Fremantle), Western Australia, Australia.

Abstract - Very well written and presents a clear summary of the proposed work to be undertaken.

Article Summary - details strengths and limitations clearly. Consideration has been given to the fact that mainly retrospective cohort studies will be identified. However, given the malignancy risks to patients an RCT comparing treatment techniques will not be conducted.

Introduction - provides clear and concise background information. The proposed study is novel as it is a meta-analysis and network meta-analysis which compare fertility treatments for women with CIN and early stage cervical cancer.

Methods and analysis:

- Objectives are clearly defined and of clinical interest to identify ways to improve patient outcomes.
- Information sources and search strategy is robust and extensive.
- Risk of bias is well considered and appropriate tools referenced.
- Statistics/Analysis are logical and comprehensive. No requirement for an external biostatistical review.
- Consumer involvement noted, this is an excellent addition to the manuscript.

Thank you for your comments.

Minor comment: Within the papers searched will you be able to determine if the patient had undergone reproductive technology prior to their CIN or early stage cervical cancer being treated?

We will record studies that include women undergoing infertility treatment and assisted reproduction. In our previous meta-analyses (Kyrgiou BMJ 2014; Kyrgiou BMJ 2016), we did not identify paper

providing separate data for women undergoing IVF. Although this will be extracted, we do not expect that a subgroup analyses will be feasible.

Reviewer: 2

Reviewer Name: N. Danhof

Institution and Country: Amsterdam UMC, The Netherlands

This manuscript represents a study protocol of a network meta analysis on local treatment for CIN, which is an important topic. To the best of my knowledge, I am not aware of a network meta analysis performed on this topic. The problem and the methods are well addressed.

Thanks for your comment.

The authors can maybe explain the fertility outcomes in more detail. In the secondary outcomes they have described total pregnancy rate. How is "pregnancy" defined?

"Total pregnancy rate" will include all possible pregnancy outcomes (miscarriage, ectopic pregnancy, molar pregnancy, termination of pregnancy, live birth, stillbirth).

This has now been amended to read:

'Total pregnancy rate, defined as any pregnancy occurring from CIN treatment till study completion irrespective of outcome (miscarriage, ectopic, molar pregnancy, termination of pregnancy, live birth, stillbirth)'

How long is the follow up time?

The follow-up in each study will be documented and reported for the time-dependent outcome measures of interest. Outcomes will be analysed for the whole study period. If there is sufficient data permitting subgroup analyses those will be analysed at prespecified intervals (ie. 6-12months).

'Total pregnancy rate will be recorded for the whole study period and/or prespecified intervals, if data is available.'

Do they extract data on time to pregnancy?

A meta-analysis explored time needed to conceive has been reported previously (Kyrgiou BMJ 2014). We have now included outcomes that incorporate time, although based on our previous analyses (Kyrgiou BMJ 2014), the data and number of studies may be limited for NMA.

'Rates of women requiring more than 12 months to conceive'

Did the authors consider to differentiate between spontaneous preterm delivery and iatrogenic preterm delivery, for example in case of pre eclampsia?

We have previously reported similar RR for total preterm birth and spontaneous preterm birth when analysed separately (Kyrgiou et al, BMJ 2016; Kyrgiou et al, Cochrane 2017). Within the context of a NMA, we selected the most clinically relevant outcomes and opted to report a single outcome including both spontaneous and iatrogenic PTB. We have added spontaneous preterm birth as an outcome, although the number of studies will be smaller and this will reduce the power of our network.

'Spontaneous PTB (sPTB), defined as less than 37 weeks of gestation'